# The prevalence of chronic kidney disease in people with severe mental illness: A systematic review protocol

**Claire Carswell** [1]*, **Kate Bramham**[2], **Joseph Chilcot**[3], **Rowena Jacobs**[4], **David Osborn**[5], **Najma Siddiqi**[1,6,7]

1 Department of Health Sciences, University of York, York, United Kingdom, 2 King's College Hospital NHS Trust, London, United Kingdom, 3 Institute of Psychiatry, Psychology and Neuroscience, King's College London, London, United Kingdom, 4 Centre for Health Economics, University of York, York, United Kingdom, 5 Division of Psychiatry, University College London, London, United Kingdom, 6 Hull York Medical School, York, United Kingdom, 7 Bradford District Care NHS Foundation Trust, Bradford, United Kingdom

* claire.carswell@york.ac.uk

## Abstract

### Background

People with severe mental illness (SMI) are more likely to develop long-term physical health conditions, including type 2 diabetes and cardiovascular disease, compared to people without SMI. This contributes to an inequality in life expectancy known as the 'mortality gap'. Chronic kidney disease (CKD) is a growing global health concern set to be the 5th leading cause of life-years lost by 2040. However, there is limited research exploring the relationship between CKD and SMI. This systematic review will aim to examine the prevalence and incidence of CKD among people with SMI.

### Methods

We will search Medline, Embase, PsycINFO, CINAHL, Scopus and Web of Science for primary epidemiological research reporting the prevalence or incidence of CKD among people with SMI in any setting. Retrieved records will be managed in Covidence and screened by two independent reviewers. Data will be extracted from included studies using a piloted data extraction form, and the quality of studies will be evaluated using the appropriate JBI Critical Appraisal Checklist. The certainty of evidence will be assessed using the Grading of Recommendations, Assessment, Development, and Evaluations (GRADE) approach. Data from the included studies will be narratively synthesised. Meta-analyses will be conducted using random effects models to examine the prevalence and incidence of CKD among people with SMI.

### Discussion

There is limited research exploring the relationship between CKD and SMI, and this proposed systematic review will be the first to examine the prevalence of CKD among people

**Data availability statement:** No datasets were generated or analysed during the current study. All relevant data from this study will be made available upon study completion.

**Funding:** Funded by an NIHR Advanced Fellowship (Award ID: NIHR303182) Overview of the award can be found here: https://fundingawards.nihr.ac.uk/award/NIHR303182 This funding is a personal Career Development Award received by CC. The funders had no role in study design, data collection and analysis, decision to publish, or preparation of the manuscript.

**Competing interests:** The authors have declared that no competing interests exist.

with SMI. This review will highlight the extent of the problem and provide a foundation for future research to improve health outcomes for people with SMI.

## Introduction

People with severe mental illness (SMI; enduring conditions that can present with psychosis, including schizophrenia and bipolar disorder) die, on average, 15–20 years earlier than people who do not have SMI [1,2]. This inequality, known as the mortality gap, is widening over time and is largely driven by higher rates of long-term physical health conditions and associated poor outcomes [1,3]. The leading cause of death among people with SMI is cardiovascular disease, accounting for 70% of all deaths in people with either bipolar disorder or schizophrenia [4]. The risk of sudden cardiac death or cardiovascular mortality is five times higher in people with SMI, compared to people without SMI [4].

While research has explored the relationship between SMI and long-term conditions such as cardiovascular disease [5] and type 2 diabetes [6], chronic kidney disease (CKD) has not received the same attention [7]. CKD is a condition characterised by progressive loss of kidney function that can eventually lead to kidney failure (CKD stage 5), requiring kidney replacement therapies such as a transplant or dialysis [8]. The current global estimate for the prevalence of CKD is 843.6 million [9]. However, prevalence is increasing, and CKD is set to become the 5th leading cause of life-years lost globally by 2040 [10].

There is evidence to suggest that people living with SMI could be at higher risk of developing CKD [7]. Antipsychotic medication and mood stabilisers, commonly used in the treatment of SMI, can induce metabolic syndrome [11], and increase the risk of diabetes mellitus (which is 2–3 times more common among people with SMI) [12] and hypertension [13] which are the leading causes of CKD worldwide [11]. For example, 27% of people with diabetes mellitus have CKD [14], while the prevalence of CKD among people with diagnosed hypertension is 27.5% [15]. Lithium, a common mood stabiliser used in the treatment of bipolar disorder, is nephrotoxic [16], can induce acute kidney injury at high doses, and increases the risk of CKD with long-term use [17]. Antipsychotic medications may also increase the risk of developing CKD [18]. Although not known to be directly nephrotoxic, this risk likely results from the significant cardiometabolic disturbances that can result as a side-effect of antipsychotic medication [19], and the well-established relationship between CKD and poor cardiometabolic health [20]. In addition, lifestyle and behavioural factors such as smoking [21], diets high in fat, salt and sugar [22], and high levels of sedentary behaviour are known risk factors for CKD, while also being more common among people with SMI [23–26].

Identifying people at risk of CKD is crucial to facilitate early identification and intervention [27]. Early intervention can prevent or slow progression to the later stages of CKD and reduce the risk of cardiovascular mortality [28]. This is pertinent as cardiovascular complications are the leading cause of death related to both CKD and SMI [29,30], yet research suggests early intervention is not occurring in this population.

People with SMI have lower rates of accessing nephrology care [31], are less likely to receive dialysis [32,33], and are less likely to be assessed for transplantation when compared to people without SMI [34]. These data are important to consider in the context of the wider health inequalities that people with SMI experience. The mortality gap and poor physical health outcomes of people with SMI are driven by varied, complex factors [35]. These factors include the issue of stigma and diagnostic overshadowing, where healthcare professionals attribute physical health symptoms to the SMI diagnosis [36], difficulties engaging

in self-management behaviours due to the symptom burden of SMI [37], the high rates of poverty, housing insecurity and social isolation among people with SMI [38], and fragmented specialism focused healthcare systems which are unable to address the interplay between physical and mental health [37,39,40].

To advocate for improved identification and appropriate care for people with co-existing SMI and CKD, there is a need to understand and describe the epidemiology of this co-morbidity, including prevalence and incidence. Therefore, a systematic review is needed to understand the extent of the problem and provide a foundation for future research.

## Objectives

This review has two overarching objectives:

- Examine the prevalence and incidence of CKD among people with SMI.

- Compare the prevalence and incidence of CKD among people with SMI, to those who do not have SMI.

## Materials and methods

This protocol has been prospectively registered on the PROSPERO database (ID: CRD42024527215) [41] and is reported in line with the Preferred Reporting Items for Systematic Reviews and Meta-Analysis Protocols (PRISMA-P) statement [42].

### Search strategy and information sources

We will search electronic databases, including Medline, Embase, PsycInfo, CINAHL, Scopus and Web of Science from conception to June 2024. No restrictions on year of publication or publication status will be applied during the initial searches, to enable exploration of prevalence and incidence across time. All searches will be re-run before the final analysis to ensure all relevant publications are included. Search strings for electronic databases were developed according to the exposure (severe mental illness), outcome (chronic kidney disease) and study design (epidemiological designs). Search terms were identified from relevant systematic reviews capturing similar concepts and were refined through initial piloting and consultation with subject librarians. The full MEDLINE strategy can be found in S1 Appendix.

We will review reference lists of included articles, and seminal publications and search key journals in the subject area to ensure all relevant publications have been identified. This will include the Clinical Journal of the American Society of Nephrology, the International Journal of Nephrology, JAMA Psychiatry, The Lancet Psychiatry, BMC Nephrology and BMC Psychiatry. We will also search grey literature repositories including ProQuest Dissertations and Open Science Framework (OSF). Experts in the field of severe mental illness and kidney disease, and authors of relevant studies, will be consulted to identify any potential key publications that could have been missed in the initial searches.

### Eligibility criteria

**Study design.**
**Inclusion criteria**

- Epidemiological observational studies, including cohort, case-control and cross-sectional studies will be included.

- Studies published in the English language

**Exclusion criteria**

- Qualitative studies, randomised controlled trials, and quasi-experimental studies will be excluded. While experimental studies may report the proportions of participants with certain conditions, they are not designed to determine the prevalence or incidence of a condition and often include super-selected samples which may not be representative of the population. Therefore, they will be excluded.

- Case reports, editorials, commentaries, and protocols will be excluded.

- Studies which are not published in the English language.

**Population/ exposure.**
**Inclusion criteria**

- Adults aged 18 years and over

- The population includes participants who have a diagnosis of SMI. SMI will be defined in this review as psychiatric conditions that can present with psychosis (not induced by substances or caused by an organic condition) [12,43–45]. These conditions include schizophrenia, schizoaffective disorder, bipolar disorder, severe depression with psychosis, other specified psychosis, and persistent delusional disorders. Any report of participants having a diagnosis of these conditions, using any standardised diagnostic tool (including the International Classification of Diseases (ICD) and the Diagnostic and Statistical Manual (DSM)) will be included.

**Exclusion criteria**

- Participants under the age of 18. If the study has also collected data from adults (over and those under 18, it will be included if the data is presented separately and can be analysed separately.

- Studies which focus exclusively on conditions which do not meet the classification for SMI, for example, anxiety disorders, depressive disorders which do not present with psychosis, eating disorders, and personality disorders, will be excluded. Studies focused on depressive disorders, in general, will only be included if they report results for severe depression with psychosis separately. Studies which focus on a variety of mental health conditions will be included if they report findings in a way that allows identification and separate analysis of participants with SMI.

- Studies, where different mental health diagnoses or categories are not reported separately (for example, where results for participants with SMI are aggregated with participants who have common mental disorders, such as depression or anxiety), will only be included if more than 50% of the sample are known to have SMI.

- Studies where the proportion of participants with SMI cannot be determined will be excluded.

**Outcome.**
**Inclusion criteria**

- Studies which report the prevalence or incidence of CKD, at any stage, among people with SMI.

- Studies which compare the prevalence or incidence of CKD, at any stage, among the general population (or those without SMI) to people with SMI.

**Exclusion criteria**

- Studies which focus on the prevalence or incidence of acute kidney injury (AKI), without reporting the prevalence or incidence of CKD, among people with SMI.

- Studies which report the prevalence of SMI among people with CKD (where a population with CKD is used as the denominator population or exposure, and SMI is reported as the outcome).

## Study selection

Records will be imported into Covidence for screening [46]. Two independent reviewers will complete the title and abstract screening, and disagreements will be resolved through discussion. If an agreement cannot be reached, the decision will be made by a third reviewer. Following title and abstract screening the full texts of eligible studies will be imported into Covidence to undergo full-text screening, which again will be completed by two independent reviewers, with disagreements resolved through discussion or a decision from a third reviewer.

## Data extraction

Two independent reviewers will perform the data extraction. A data extraction form will be developed in Excel and initially piloted by the two reviewers on a subset of included studies to ensure the form is fit for purpose. Disagreements and inconsistencies in data extraction will be reviewed and discussed, and the initial data extraction form will be refined as needed. Following the piloting, data extraction will be performed again by two reviewers, with any disagreements being resolved by a third reviewer who has not previously been involved with the data extraction process. Articles reporting data from the same study in different publications will be grouped at this stage and presented within a single study. The data extraction form will capture the author's names, date of publication(s), publication type(s), country, study design, total sample size, number of participants with SMI, diagnoses, mean age, percentage of females, ethnicity, medication use, data collection procedures, statistical approach, summary results (prevalence, incidence and comparisons), and covariates considered.

## Risk of bias assessment

Two independent reviewers will use the appropriate JBI (formerly Joanna Briggs Institute) Critical Appraisal Checklist to evaluate the risk of bias in each of the included studies. The JBI Critical Appraisal Checklists assess the trustworthiness and quality of published research and include checklists for a range of different methodologies including cohort studies, case-control studies and cross-sectional studies [47]. Any disagreement in the assessment of the two independent reviewers will be settled through discussion, or if an agreement cannot be reached, through a decision made by a third reviewer who has not previously been involved with the critical appraisal process.

## Certainty of evidence

We will use the Grading of Recommendations, Assessment, Development, and Evaluations (GRADE) approach to determine the certainty of the evidence [48]. GRADE outlines five domains that are assessed to determine certainty: risk of bias, inconsistency, indirectness, imprecision and publication bias. GRADE is predominantly used to assess certainty in intervention research and has not been formally adapted for the use in systematic reviews of prevalence or incidence. However, previous systematic reviews have used GRADE to evaluate the certainty of prevalence studies with adapted components [49]. We will use the GRADE guidance for baseline risk or overall prognosis, as recommended by Borges Migliavaca et al. (2020) for systematic reviews of prevalence [50].

## Data synthesis

The data from the included studies will be presented in tables and narratively synthesised to provide a summary of the study characteristics and findings. Meta-analyses will also be conducted using random effects models assuming an adequate number of studies (at least two) and clinical similarity (e.g., prevalence/incidence measures including similar stages of CKD and methods of staging CKD) to justify the pooling of prevalence or incidence rate. The heterogeneity across studies will be assessed through visual inspection of forest plots, a chi-squared test for heterogeneity, and the $I^2$ statistic. However, as meta-analyses of prevalence tend to have high $I^2$ statistics [50], we will not apply statistical thresholds to determine whether it is appropriate to pool the data. We will transform the raw proportions using the Freeman-Tukey variant of the arcsine square root transformation [47]. For objective 1, the pooled prevalence or incidence rate of CKD among people with SMI will be presented with 95% CIs. For objective 2, we will undertake a meta-analysis of prevalence or incidence rate comparisons if the data is available. These will be presented as pooled risk ratios or odds ratios, with 95% CIs. If there is sufficient data, we will conduct subgroup analyses to explore the differences in CKD across different types of SMI diagnosis, medication, gender, age, geography (Low- and middle- income countries or high-income countries), setting (community or inpatient), and year of data collection.

## Discussion

This proposed review aims to address a crucial gap in the evidence base by synthesising the literature on the prevalence of CKD among people with SMI. Previous systematic reviews have established that people with SMI are at significantly higher risk of developing long-term physical health conditions, including type 2 diabetes [5,6]. However, the available literature on the prevalence or incidence of CKD has not yet been synthesised in a systematic review.

While this review aims to address an important gap, the conclusions it will be able to draw are limited due to the nature of epidemiological research. This review will describe the current literature on the prevalence of CKD among people with SMI, but definitions, diagnostic criteria, trends and patterns of disease change over time and across populations and therefore the findings may be highly heterogenous. This review will aim to describe those differences and pool data only where appropriate and relevant. Additionally, we are planning to exclude studies that are not published in the English language, which may further limit the generalisability of the findings.

### Implications for research, practice or policy

By examining the prevalence of CKD in people with SMI, this review will highlight the need for future research and policy change. Most research to date in this population has focused on the role of lithium as a risk factor for CKD [51], and there is limited research exploring other, potentially modifiable, risk factors [7]. Establishing the prevalence of CKD among people with SMI (including those not on lithium treatment) is an important step towards identifying key modifiable risk factors that contribute to the development and progression of CKD in this population. Additionally, depending on the available research, we may be able to describe differences in CKD prevalence over time or across different populations, identify potential inequalities that could inform clinical practice. In summary, this will enable not just more targeted approaches to screening and monitoring to facilitate early intervention but could also inform the development of future interventions to improve CKD outcomes for people with SMI.

## Supplementary information

**S1 Appendix. MEDLINE Search Strategy.**
(DOCX)

**S1 Checklist. PRISMA-P-SystRev-checklist.**
(DOCX)

## Author contributions

**Conceptualization:** Claire Carswell, Kate Bramham, Joseph Chilcot, Rowena Jacobs, David Osborn, Najma Siddiqi.

**Funding acquisition:** Claire Carswell, Kate Bramham, Joseph Chilcot, Rowena Jacobs, David Osborn, Najma Siddiqi.

**Investigation:** Claire Carswell.

**Methodology:** Claire Carswell, Kate Bramham, Joseph Chilcot, Rowena Jacobs, David Osborn, Najma Siddiqi.

**Project administration:** Claire Carswell.

**Supervision:** Kate Bramham, Joseph Chilcot, Rowena Jacobs, David Osborn, Najma Siddiqi.

**Writing – original draft:** Claire Carswell, Kate Bramham, Joseph Chilcot, Rowena Jacobs, David Osborn, Najma Siddiqi.

**Writing – review & editing:** Claire Carswell, Kate Bramham, Joseph Chilcot, Rowena Jacobs, David Osborn, Najma Siddiqi.

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
