## [Decision Letter · Decision Letter 0]

20 Nov 2024

PONE-D-24-38248The prevalence of chronic kidney disease in people with severe mental illness: A systematic review protocolPLOS ONE

Dear Dr. Carswell,

Thank you for submitting your manuscript to PLOS ONE. After careful consideration, we feel that it has merit but does not fully meet PLOS ONE’s publication criteria as it currently stands. Therefore, we invite you to submit a revised version of the manuscript that addresses the points raised during the review process.

We look forward to receiving your revised manuscript.

Kind regards,

Eleni Petkari

Academic Editor

PLOS ONE

Journal Requirements: When submitting your revision, we need you to address these additional requirements. 1. Please ensure that your manuscript meets PLOS ONE's style requirements, including those for file naming. The PLOS ONE style templates can be found at https://journals.plos.org/plosone/s/file?id=wjVg/PLOSOne_formatting_sample_main_body.pdf and https://journals.plos.org/plosone/s/file?id=ba62/PLOSOne_formatting_sample_title_authors_affiliations.pdf

Reviewers' comments:

Reviewer's Responses to Questions

**Comments to the Author**

1. Does the manuscript provide a valid rationale for the proposed study, with clearly identified and justified research questions?

Reviewer #1: Yes

Reviewer #2: Yes

2. Is the protocol technically sound and planned in a manner that will lead to a meaningful outcome and allow testing the stated hypotheses?

Reviewer #1: Yes

Reviewer #2: Partly

3. Is the methodology feasible and described in sufficient detail to allow the work to be replicable?

Reviewer #1: Yes

Reviewer #2: Yes

4. Have the authors described where all data underlying the findings will be made available when the study is complete?

Reviewer #1: Yes

Reviewer #2: Yes

5. Is the manuscript presented in an intelligible fashion and written in standard English?

Reviewer #1: Yes

Reviewer #2: Yes

6. Review Comments to the Author

You may also provide optional suggestions and comments to authors that they might find helpful in planning their study.

Reviewer #1: Introduction: The introduction would benefit from a more in-depth literature review that incorporates similar studies, especially those examining the prevalence of CKD in populations with comorbidities. This will strengthen the context of your study and demonstrate how it builds upon or fills gaps in existing research.

Justification: It would be helpful to provide a stronger justification for why this systematic review is necessary. While the introduction briefly touches on the mortality gap and the potential higher risk of CKD in individuals with severe mental illness (SMI), expanding on why this population is underserved in research would better emphasize the importance of this review.

Potential Limitations: I recommend expanding the discussion around the potential limitations of the study. This could include challenges related to the heterogeneity of the studies included in the meta-analysis, variations in diagnostic criteria for CKD and SMI, and the potential exclusion of unpublished data or data from grey literature.

Language Restriction: It is unclear whether only studies published in English will be included in the review. If this is the case, it is important to specify this limitation and discuss how it may impact the generalizability of the findings.

Discussion Section: The discussion section could be expanded to further elaborate on how the findings will contribute to clinical practice, policy, or future research. This would help to better contextualize the significance of your review's results beyond the immediate field of study.

Changes in CKD Staging: There have been recent updates in the staging of chronic kidney disease. It would be beneficial to address how these changes will affect your systematic review, particularly regarding the inclusion of older studies that may have used previous CKD classification systems.

Reviewer #2: The topic itself is unique. However, please explain if systematic review is the most feasible method for assessing prevalence and incidence rates even if goal is to measure at a global level.

Secondly, please explain what is meant by "narrative synthesis" will be done for data retrieved from the included studies.

Thirdly, explain how meta-analysis is most suitable for conducting a study on population trends.

Fourthly, how would you ensure homogeneity of study sample by including ProQuest Dissertations and Open Science Framework (OSF) in addition to publications from databases.

Fifthly, also justify how can just assessing prevalence and incidence rates in patients with mental health illnesses be directly related with the health outcomes improvement in the targeted population.

7. PLOS authors have the option to publish the peer review history of their article (what does this mean? ). If published, this will include your full peer review and any attached files.

**Do you want your identity to be public for this peer review?** For information about this choice, including consent withdrawal, please see our Privacy Policy .

Reviewer #1: **Yes: ** Ang Yee Gary

Reviewer #2: **Yes: ** Dr. Saira Akhlaq

---

## [Editor Report · Decision Letter 1]

8 Jan 2025

The prevalence of chronic kidney disease in people with severe mental illness: A systematic review protocol

PONE-D-24-38248R1

Dear Dr. Carswell,

We’re pleased to inform you that your manuscript has been judged scientifically suitable for publication and will be formally accepted for publication once it meets all outstanding technical requirements.

Kind regards,

Eleni Petkari

Academic Editor

PLOS ONE
---

## [Editor Report · Acceptance letter]

PONE-D-24-38248R1

PLOS ONE

Dear Dr. Carswell,

I'm pleased to inform you that your manuscript has been deemed suitable for publication in PLOS ONE. Congratulations! Your manuscript is now being handed over to our production team.

Kind regards,

on behalf of

Dr. Eleni Petkari

Academic Editor

PLOS ONE